# The Therapeutic Potential of FLASH-RT for Pancreatic Cancer

**DOI:** 10.3390/cancers14051167

**Published:** 2022-02-24

**Authors:** Chidi M. Okoro, Emil Schüler, Cullen M. Taniguchi

**Affiliations:** 1Department of Experimental Radiation Oncology, The University of Texas MD Anderson Cancer Center, Houston, TX 77030, USA; cmokoro@mdanderson.org; 2Department of Radiation Oncology, The University of Texas MD Anderson Cancer Center, Houston, TX 77030, USA; 3Department of Radiation Physics, The University of Texas MD Anderson Cancer Center, Houston, TX 77030, USA

**Keywords:** FLASH radiation, pancreatic cancer, conventional radiation, FLASH effect, gastrointestinal toxicity

## Abstract

**Simple Summary:**

Ultra-high dose rate radiation, widely nicknamed FLASH-RT, kills tumors without significantly damaging nearby normal tissues. This selective sparing of normal tissue by FLASH-RT tissue is called the FLASH effect. This review explores some of the proposed mechanisms of the FLASH effect and the current data that might support its use in pancreatic cancer. Since radiation for pancreatic cancer treatment is limited by GI toxicity issues and is a disease with one of the lowest five-year survival rates, FLASH-RT could have a large impact in the treatment of this disease with further study.

**Abstract:**

Recent preclinical evidence has shown that ionizing radiation given at an ultra-high dose rate (UHDR), also known as FLASH radiation therapy (FLASH-RT), can selectively reduce radiation injury to normal tissue while remaining isoeffective to conventional radiation therapy (CONV-RT) with respect to tumor killing. Unresectable pancreatic cancer is challenging to control without ablative doses of radiation, but this is difficult to achieve without significant gastrointestinal toxicity. In this review article, we explore the propsed mechanisms of FLASH-RT and its tissue-sparing effect, as well as its relevance and suitability for the treatment of pancreatic cancer. We also briefly discuss the challenges with regard to dosimetry, dose rate, and fractionation for using FLASH-RT to treat this disease.

## 1. Introduction

External beam radiation therapy uses ionizing radiation to induce double strand DNA breaks in cancer cells and is thought to selectively kill tumors by exploiting differences in DNA repair between tumors and normal tissues. When radiation therapy is administered to patients, image guidance is used to guide the precise targeting of tumors but cannot eliminate incidental radiation injury to normal tissues. Thus, the doses of radiation used in modern oncology have generally been derived empirically, usually as the maximum permitted by normal tissue toxicity [1]. For some forms of cancer— e.g., early-stage cancers of the lung, prostate, anus, head and neck, and cervix—radiation given with radiosensitizing chemotherapy is sufficient for a cure [2,3,4,5,6]. However, many tumors in the abdomen cannot be sufficiently treated with only radiation and chemotherapy, since the maximum dose of radiation that the intestines can receive over a lifetime is about 50 Gy [7]. Unfortunately, many tumors of the abdomen require much more than 50 Gy to be completely sterilized. Therefore, radiation can be a useful preparative treatment before surgery, but it is often insufficient to achieve oncologic control on its own. Surgery is limited as a treatment option and poses risks such as pain and infection. Chemotherapy shares similar risks of pain as surgery and improvements are limited. Without advancements in radiation protection or radiosensitization, the curative potential of radiation therapy alone for gastrointestinal tumors will be limited.

## 2. FLASH-RT and the FLASH Effect

FLASH-RT represents an entirely new paradigm of potentially curative therapy that is broadly applicable across all cancer types. The ability to treat tumors while sparing normal tissues is essentially the ‘holy grail’ of cancer therapy since all oncologic therapies are limited by patient toxicity (chemotherapy and radiation side effects or morbidity from surgery).

FLASH-RT encompasses two components: (1) The delivery of ultra-high dose rates (UHDR) and (2) the radiation sparing effect (a.k.a. the FLASH effect). Regarding the former, FLASH-RT is a catchy nickname for the novel radiation treatment that uses ultra-high dose rates to deliver radiation therapy (RT). In conventional radiation therapy (CONV-RT) used in most of the world, the dose rate is generally <0.1 Gy/s, whereas the FLASH-RT dose rate is generally defined as a mean dose rate of ≥40 Gy/s [8,9]. A more precise definition is still being debated and will be important to define in order to ensure scientific rigor and to safeguard the clinical translation of FLASH-RT. Moreover, a more mature definition of FLASH-RT should, at a minimum, include other interdependent parameters, such as dose, dose per pulse, instantaneous dose rate, and total duration of exposure [9,10]. 

Beyond the ultra high dose rates, the ability for FLASH-RT to selectively spare normal tissues while maintaining tumor killing is what is generating excitement for this new technique. This novel biological phenomenon of selective normal tissue sparing is known as the FLASH effect [11]. The FLASH effect has now been shown repeatedly in many different organ systems, with a dose modifying factor of 1.1–1.8, depending on model species and end point used [11]. However, the mechanism of the FLASH effect has yet to be fully elucidated but is a critical component in the understanding and optimization of FLASH-RT [12]. It is critical to note that the FLASH effect (normal tissue sparing with isoeffective killing of tumors) is what defines FLASH-RT, and thus future trials testing this modality will focus on reducing toxicity. 

## 3. Mechanisms of the FLASH Effect

### 3.1. Differential DNA Damage

Differences in the extent and type of DNA damage resulting from FLASH- versus CONV-RT could contribute to the observed FLASH effect. The highly energetic ionizing radiation used in-RT is what kill tumors. Specifically, the energy of this type of radiation induces double-stranded DNA breaks that overwhelm the cell’s repair mechanism, leading to cell death. FLASH- and CONV-RT seem to cause different types of DNA damage in normal tissues and tumors [13]. A FLASH-RT dose of 20 Gy has been shown to cause less double-stranded DNA damage to normal cells than CONV radiation does at the same dose [14]. In addition, although the time required for DNA damage repair is similar for CONV radiation and FLASH-RT, UHDR has been shown to produce fewer dicentric chromosomes [14]. Whether the damage that FLASH-RT inflicts on tumor cells differs significantly from CONV-RT remains unknown. Nevertheless, the lesser DNA damage to normal tissue from FLASH-RT than from CONV-RT may partially explain the FLASH effect. 

### 3.2. Oxygen Depletion Hypothesis

Oxygen tension is a critical regulator of radiation sensitivity in classical radiobiological models. Higher oxygen levels have correlated with increased DNA damage, sometimes explained by the ‘oxygen fixation hypothesis’, in which damage induced by radicals is ‘fixed’ by molecular oxygen, and thus cannot be chemically restored. Oxygen depletion has known radioprotective effects on both normal and tumor tissues [15].

Consequently, one of the most popular proposed mechanisms of the normal tissue sparing induced by FLASH-RT is known as the oxygen depletion or transient hypoxia hypothesis. Ionizing radiation hydrolyzes cytoplasmic water to yield free radicals that can ultimately cause permanent DNA damage [15]. The processes of downstream formation of radical species and damage fixation are processes that consume oxygen. Thus, a lack of oxygen limits the extent of radiation-induced damage. According to the oxygen depletion hypothesis, the UHDR irradiation creates a temporarily hypoxic environment that confers transient radioresistance in the irradiated tissue (Figure 1) [16,17]. Furthermore, the time interval of delivery (<100–200 ms) is short enough to negate the effect of reoxygenation during the radiation delivery [18]. The phenomenon of transient hypoxia would not occur in CONV-RT because the significantly lower dose rates involved would allow reoxygenation to occur during delivery of the radiation.

To achieve the necessary reduction to hypoxic levels without excessive doses of radiation (assuming an oxygen depletion rate of around 0.5 mmHg/Gy) [19], the cells would need to be in physiological, or even close to hypoxic levels, of oxygenation prior to irradiation. Although this not the case for most tissues, some niches of stem cells reside in hypoxic microenvironments, with partial oxygen concentrations as low as 10 mmHg in many different tissue types [20]. The differential effect of normal vs. tumor tissue (normal tissue sparing but iso-effective response on tumor tissue) would then be explained by the already highly hypoxic tumor and would therefore not be affected. Abolfath et al. demonstrates in simulations the significance of different oxygen levels between tumors and normal tissues on the induction of the FLASH effect [21].

However, the oxygen depletion hypothesis has been put into question, for several reasons. First, most solid tumors are known to not be uniformly hypoxic, having both hypoxic and non-hypoxic compartments throughout the tumor [22]. Under the transient hypoxia hypothesis, this heterogeneity should then introduce a sparing effect also on tumors, which is not supported by the literature [9]. Second, in vitro studies of normal, non-immortalized cells have shown a sparing effect when irradiated in ambient air conditions (pO2 ≈ 159 mmHg) [23,24,25,26]. Third, simulation studies and oxygen consumption studies in pure water and cell buffer solutions have also challenged this hypothesis by showing that total oxygen depletion did not occur with the use of FLASH radiation [27,28,29]. However, whether total depletion of oxygen is required for the induction of the FLASH effect has yet to be confirmed.

Independent of the magnitude of the effect that radiological reduction in oxygen concentration has on tissues after UHDR radiation, oxygen most likely still underlies the FLASH effect. Modulation of the partial oxygen pressure in in vitro experiments had an important effect on the resulting survival fraction and in the increased sparing effect seen after FLASH relative to CONV irradiation [30,31,32]. Similar results have been seen in vivo when increasing the oxygen tension of tissues in mice by having the mice breathe carbogen before and during irradiation, which led to no differences in effects after FLASH versus CONV irradiation [33]. Mice breathing medical air, by contrast, showed strong and reproducible FLASH effects.

The role of oxygen is therefore potentially better explained by radical–radical interactions, where high local concentrations of radicals being formed during UHDR irradiation cause the radicals to interact between themselves and effectively ‘quench’ the downstream effects of the produced radicals. Current models on the lifetime of radical formation support this hypothesis. In their models, Labarbe et al. showed the importance of oxygenation level and dose rate in the formation and lifetime of organic peroxyl radicals [34]. Interestingly, moderate oxygenation levels resulted in reduced lifetime of organic peroxyl radicals following UHDR irradiation, while no observed differences were found at low or high oxygenation levels. Other simulation studies on the FLASH effect and its relationship to intracellular oxygen are related to the redox biology specific to normal and tumor tissue [35]. Tumor tissue has an increased ‘normal’ level of reactive oxygen species (ROS) and also has a decreased capacity to handle a rapid increase in ROS levels. This near-saturated system of ROS handling, together with higher levels of labile iron and transferrin receptors in tumor compared to normal tissue, could result in higher induced oxidative damage. Some of these hypotheses are directly testable in an experimental setting, which would be needed to fully elucidate the role of oxygen in relation to the FLASH effect.

**Figure 1 cancers-14-01167-f001:**
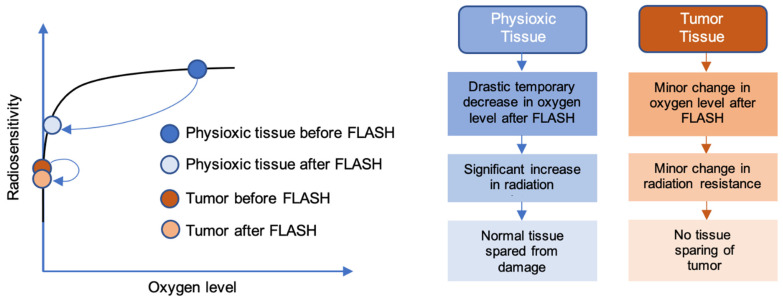
Oxygen depletion hypothesis of the FLASH effect. Tumors are extremely hypoxic relative to their surrounding normal tissue, with oxygen tensions ranging from 0.3% to 4.2%. The surrounding normal tissue is more highly oxygenated (i.e., physioxia), usually between 3% and 7.4% [36]. For this reason, a more pronounced decrease in oxygen level after FLASH treatment is observed in physioxic conditions compared with tumors. This correlates with a significant decrease in radiosensitivity (or increase in radiation resistance) in normal tissues, conferring the FLASH effect. Tumors, on the other hand, show little change in radiosensitivity, leading to the absence of the FLASH effect.

### 3.3. Immune Sparing Hypothesis

Lymphocytes are among the most radiosensitive tissues and are quickly depleted with prolonged radiation exposure [37]. However, the degree of radiosensitivity of lymphocytes differs depending on whether they are within tumors or are circulating. A recent mouse model study demonstrated that lymphocytes residing in tumors are more radioresistant than are circulating lymphocytes [38]. Specifically, as much as 85% of these resident lymphocytes were preserved compared to 10% of circulating lymphocytes after exposure to 8 Gy whole body irradiation. The same study showed that TGFβ was a key contributor to the increased radioresistance exhibited by tumor-infiltrating lymphocytes. Levels of circulating tumor cells are therefore good markers of the extent of immunocompromise resulting from radiation.

This brings us to another popular hypothesis involving modulation of the immune system, known to be crucial in tumor biology and treatment responses. Isolated lymphopenia is practically a pathognomonic feature of conventional irradiation of large areas of the body [38]. FLASH-RT is thought to reduce lymphopenia by decreasing the volume of blood irradiated in any body part (Figure 2). As a hypothetical example, exposure of a large mediastinal radiation field to 4 Gy would entail irradiation of not only many lymph nodes, but also the heart and pulmonary vessels, which oxygenate and recirculate the entire blood volume at a rate of 5 L/min [39]. This modest 4-Gy dose could be delivered with CONV-RT at a typical dose rate of 0.08 Gy/s in 50 s, but even such a modest dose would not only irradiate the mediastinal lymph nodes but also approximately 4L of blood circulating throughout the irradiated volume. In this simple example, CONV-RT would expose nearly the entire circulating blood volume, and radiosensitive lymphocytes in particular, to some dose of radiation. On the other hand, FLASH-RT at dose rates of 40 Gy/s and above would involve exposing only a fraction of the blood volume to radiation, thereby effectively limiting the total exposure of lymphocytes to potentially lethal radiation injury.

This hypothetical scenario was supported by a recent mouse simulation study in which exposure to CONV-RT killed 90% to 100% of circulating immune cells as compared with FLASH-RT, which killed only 5% to 10% of such cells in the same treatment setup [40]. That same study showed that immune cell sparing in mice was optimal at FLASH-RT dose rates of at least 40 Gy/s. In contrast, an experimental study showed that radiation delivered at 30 Gy/s failed to protect mice from cardiac and splenic radiation-induced lymphopenia [41]. However, the dose rate of 30 Gy/s may not have been sufficient to induce the FLASH effect in this study. Nevertheless, even at this dose rate, the recruitment of T lymphocytes into the tumor microenvironment was more prevalent after FLASH-RT than after CONV-RT [41]. More studies are needed to better elucidate the contribution of the immune response on the mechanism underlying the FLASH effect.

## 4. Preclinical Models of FLASH-RT

There are several studies showing the promising tissue sparing effects of FLASH-RT. Most have focused on rodent experiments where it has been shown that FLASH-RT induces tissue sparing effects in, e.g., GI tract (crypt regeneration, survival) [42], brain (cognition, neuroinflammation) [43,44], and skin (early skin reactions, necrosis, survival) [45,46,47], compared to CONV-RT. Utilizing rodent models, it has also been shown that the higher therapeutic index of FLASH compared to CONV-RT is due to the reduction in normal tissue toxicity, as the tumor response has been shown to be independent of mode of delivery [48,49]. For instance, one preclinical orthotopic model demonstrated the effectiveness of FLASH-RT in preserving intestinal stem cells to the same degree as in healthy mice, while showing an isoeffective treatment on the ID8 ovarian tumor model compared to CONV irradiation [50].

In large animal studies, FLASH-RT was seen to result in lower occurrences of fibro-necrosis in pig skin compared to CONV-RT [51]. In cats with locally advanced squamous cell carcinoma, little to no acute toxicity was found with FLASH-RT [51]. However, it should be noted that other studies using radiation therapy roughly equivalent to FLASH-RT showed no difference in tumor control and no significant tissue sparing [52].

## 5. Clinical Experiences with FLASH-RT

To date, only one patient has been treated with FLASH-RT from which the results have been made available to the scientific community. In this case, FLASH-RT was shown to be safe. This patient, a 75 year old male with cutaneous lymphoma, was the first to be treated with FLASH-RT [53]. He had previously underwent localized CONV-RT to treat painful cutaneous lesions but exhibited poor tolerance. FLASH-RT was suggested as a possible radiation treatment that would provide similar tumor control with less toxicity. Treatment with FLASH-RT involved irradiating a 3.5-cm diameter skin tumor on the patient with a 5.6-MeV LINAC capable of delivering FLASH-RT dose rates. The results of the treatment were favorable where the patient exhibited no decrease in skin thickness and an equivalent tumor control compared to previously administered CONV-RT treatments [53]. In a different study on the same patient, two distinct tumors were treated to a dose of 15 Gy single fraction using either CONV or FLASH dose rates [54]. Both treatments gave comparable results in terms of acute and late effects as well as tumor control.

Human clinical trials that aim to access the feasibility of FLASH-RT are already underway at the time of this review (February 2022). For instance, Cincinnati Children’s/University of Cinncinati Health Proton Therapy Center initiated a prospective trial under the name FAST-01. The trial began in November 2020 and aims to enroll 10 patients with bone metastases. The main objective of this trial is to determine the feasibility of 1 fraction of 8 Gy FLASH-RT using protons for human radiation treatments [55]. Centre Hospitalier Universitaire Vaudois (CHUV), Lausanne University Hospital, which performed the first human treatment using FLASH-RT, began enrollment for a new FLASH-RT clinical trial in June 2021. Their phase I trial, set to enroll 7 to 21 patients, aims to determine an optimal FLASH-RT dose for improved tumor control in melanoma skin metastases while preventing radiation-induced toxicity [56].

Thus, overall, this new technology has been used in humans in a limited capacity, but appears to be safe awaiting further clinical trial data. Based on this promise, we posit potential addition of usage in the setting of pancreatic cancer.

## 6. Pancreatic Ductal Adenocarcinoma: An Ideal Use for FLASH?

The use of FLASH-RT to administer ablative doses of-RT, and therefore the potential for curative therapy, addresses a critical unmet need for patients with unresectable pancreatic cancer. The potential for killing tumors while minimizing toxicity would be perfectly applied to pancreatic cancer, where growth of the primary tumor causes pain and suffering for nearly all afflicted individuals. If FLASH-RT meets even a fraction of its potential in this disease, that would improve the quality of life for patients undergoing treatment and for survivors. With a survival rate of less than 10% at 5 years, even a modest improvement in disease outcomes could double or triple survival rates [57]. Surgery remains the only curative form of treatment, but surgery is an option for only about 10–15% of patients, because most pancreatic cancers present with invasion into nearby blood vessels. Radiation can sometimes be used when surgery is not possible, but its efficacy is limited by the anatomic location of the pancreas (Figure 3a). Moreover, pancreatic cancer is highly resistant to radiation therapy, and therefore ablative doses exceeding 70 Gy are needed for effective tumor control [58]. Unfortunately, the pancreas abuts the duodenum, stomach, and small bowel, which can only tolerate about 50 Gy of radiation. As a result, the dose of radiation that are given to patients with pancreatic cancer is safe but not sufficient to permanently control the disease.

Why is targeting the pancreas so difficult? Radiation therapy targets a focused three-dimensional area that must always include a rim of normal tissue (roughly 2–3 mm) to account for changes in patient position and movement of internal organs. Although the pancreas is generally not considered to be mobile within the body, its position can change by as much as 2 cm just from normal respiration. Changes in bowel gas and peristalsis can also alter the amount of normal tissue that happens to be in a radiation field on a given day. Thus, eliminating all normal tissue from a radiation field is not realistic.

FLASH-RT is an ideal solution to targeting pancreatic cancer since it can also target the unique biology of pancreatic cancer. A common characteristic of most tumors is a low level of oxygen known as hypoxia. Pancreatic ductal adenocarcinoma (PDAC) is the most common form of pancreatic cancer, marked by a highly hypoxic tumor environment [59], which makes it resistant to both chemotherapy and radiation [60]. Notably, PDAC tumors are extremely hypoxic compared to their surrounding healthy tissue under physioxia (Figure 3b). FLASH-RT may perfectly enable effective hypoxic tumor killing while sparing the healthy intestinal tissue nearby (Figure 3c). This is because the highly hypoxic tumors of PDAC will allow for a more pronounced transient oxygen depletion, enhancing the FLASH effect. Moreover, FLASH-RT would also reduce lymphopenia, which would enable further systemic treatments such as chemotherapy and immunotherapy [61].

FLASH-RT would be a completely new weapon against pancreatic cancer because it allows treatments that were previously not possible. For instance, although most ablative techniques target the tumor and a small rim of normal tissue, FLASH-RT could enable elective irradiation of at-risk lymph nodes and at-risk organs that may harbor microscopic disease, such as the liver. FLASH-RT could also be combined with other therapies to further improve its therapeutic ratio. Our group exploited this concept by using selective hypoxia mimicry in normal tissue [62], or tumor-specific radiosensitization [63], to improve the therapeutic ratio of ablative radiation in pancreatic cancer. Finally, existing clinical LINACs could be retrofitted to allow FLASH-RT capability [64], which would enable this type of treatment to be given in many parts of the world that lack new, state-of-the-art radiation delivery machines, which may enhance the possibility of curative treatment.

## 7. Challenges to Translation

Most FLASH-RT experiments are performed with specialized machines or clinical LINACs that have been converted to produce ionizing radiation at ultra-high dose rates. The generally accepted threshold dose rate of 40 Gy/s is thought to be needed to induce the FLASH effect [65]. Even if a machine could produce this dose rate, there is the issue with properly calibrating the accuracy of the dose delivered. There have been several solutions to this issue of dosimetry of FLASH, which will likely be an important component going forward, particularly for deep tumors seen in pancreatic cancer.

Currently, IntraOp Medical has developed an electron-based FLASH LINAC for intraoperative radiation. It has recently been approved for FLASH-RT preclinical experiments and clinical human trials [66]. The use of intraoperative radiation (IORT) in pancreatic cancer has shown prior benefit in local control after resection since it helps to reduce microscopic disease near the retroperitoneal border, which is often challenging to achieve clear margins [67]. However, resectable pancreatic cancer may not be the best first indication for FLASH-RT in pancreatic cancer, since most concerning normal tissue can be manually retracted by surgery during IORT, therefore reducing the need for the sparing effect of FLASH-RT. An electron FLASH approach might be useful in the unresectable setting, particularly if a laparotomy is performed first to expose the intact tumor.

Another limitation of studies of FLASH-RT to date has been their use solely as single fractions. On the one hand, fractionation may not be needed in many cases given the apparent normal-tissue-sparing effects of FLASH-RT. On the other hand, however, fractionation may be needed to deliver a fully ablative dose to radiation-resistant tumors such as pancreatic cancer. Stereotactic body radiation therapy (SBRT), in which radiation is given in 3–5 fractions, has been shown to improve outcomes while reducing toxicity through its more accurate targeting [68,69,70]. Whether the FLASH effect is still present if the radiation is given in a fractionated fashion is currently unknown. Therefore, FLASH dose-finding experiments such as these must be conducted in preclinical settings to determine their safety, efficacy, and feasibility before moving on to clinical trials.

## 8. Conclusions

FLASH-RT is a relatively novel form of radiation therapy that involves the use of ionizing radiation at UHDRs, which distinguishes FLASH-RT from CONV-RT, which involves much lower dose rates. Preclinical studies have shown that FLASH-RT can confer healthy tissue sparing through a phenomenon known as the FLASH effect. The mechanism of the FLASH effect has yet to be fully elucidated, but several hypotheses have been proposed as outlined in this review. Although some contradictions to these hypotheses have been noted, some combination of them—along with other unknown factors—probably participate in creating the FLASH effect. Furthermore, in order to safely bring FLASH-RT to the clinic, the precise definition of what constitutes FLASH-RT is urgently needed.

The FLASH effect may be ideally suited for the treatment of pancreatic cancer for several reasons. Pancreatic cancer is characterized by a highly hypoxic tumor environment. Surgery remains the only curative option for localized pancreatic cancer, with radiation often used as an adjuvant after surgery. However, the pancreas’ proximity to the gut and critical blood vessels precludes use of ablative radiation doses in most cases. However, the FLASH effect could help to minimize damage to these normal tissues, which would enable high radiation doses to be administered. FLASH-RT has shown promise in several mouse models and in a few anecdotal clinical situations, but more rigorous study is needed to determine its role in the treatment of pancreatic cancer. The recent expansion of FLASH-RT using intraoperative RT (IORT) equipment opens up the potential for future human trials targeting pancreatic cancer with IORT techniques. This would greatly expand the opportunities for testing FLASH-RT and for further exploring its potential. Considering the relatively low 5-year survival rates for patients with pancreatic cancer, advances in FLASH-RT may prove to be pivotal in significantly improving treatment outcomes in this difficult-to-treat form of cancer.

## Figures and Tables

**Figure 2 cancers-14-01167-f002:**
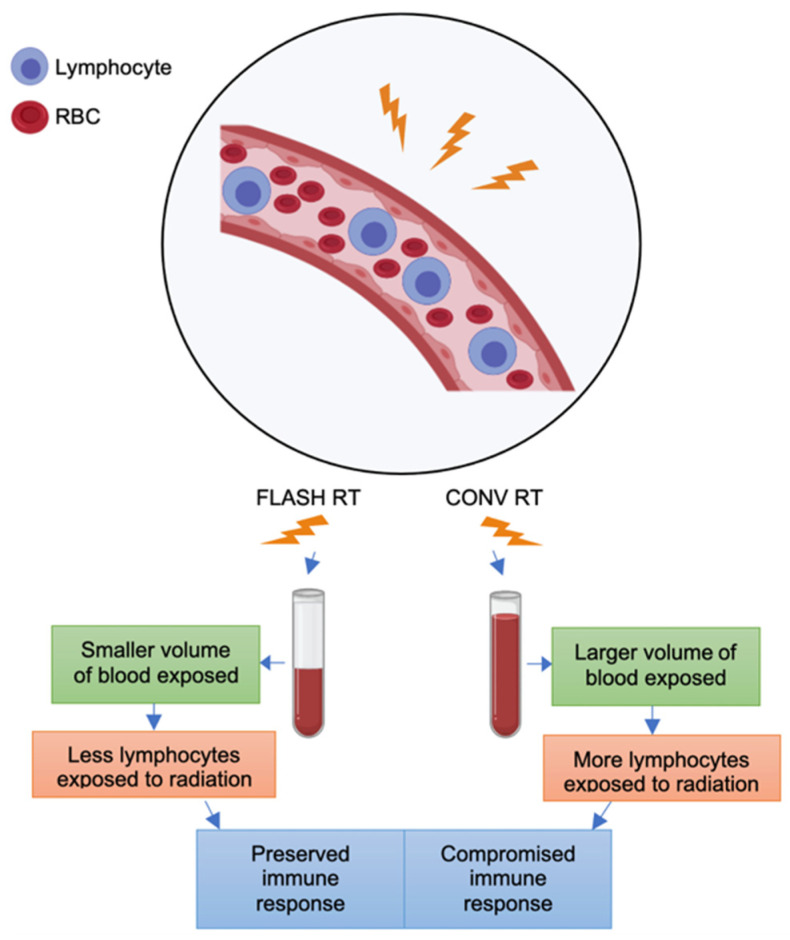
Immune hypothesis for the FLASH effect. Lymphocytes circulating within blood vessels are an important component of the immune response that influences tumor suppression. According to the immune hypothesis, the higher dose rates characteristic of FLASH-RT allow exposure of a much smaller volume of blood to radiation than CONV-RT. As a result, a higher number of circulating lymphocytes will survive and the immune response critical for tumor suppression is preserved to a greater degree. In contrast, the lower dose rates used in CONV-RT allow exposure of larger blood volumes circulating through the radiation field, leading to a significant loss of lymphocytes and a compromised immune response.

**Figure 3 cancers-14-01167-f003:**
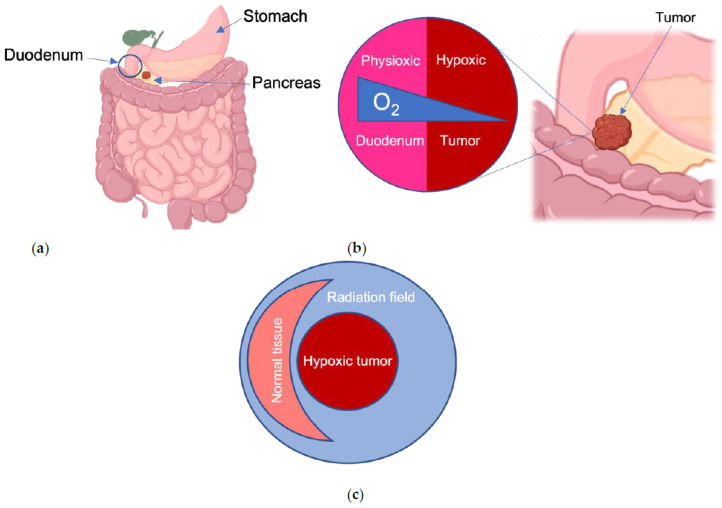
Gastrointestinal organs are susceptible to radiation damage from pancreatic cancer radiotherapy. (**a**) The duodenal portion of the small intestine, the stomach, and other nearby organs are prone to radiation-induced toxicity, especially the rapidly dividing intestinal cells; (**b**) Oxygen concentrations (blue wedge) vary between normal tissues and tumors. Pancreatic tumors near the duodenum are especially hypoxic relative to the physioxic environment of the healthy tissue at risk; (**c**) Nearby normal tissues that would be damaged by CONV-RT would be spared by FLASH-RT, and FLASH-RT would also kill the tumor through the FLASH effect.

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
