# Peer review of "The Therapeutic Potential of FLASH-RT for Pancreatic Cancer"

_cancers, 2022, doi:10.3390/cancers14051167_

Round 1

Reviewer 1 Report

The review describes very promising FLASH RT targeted method to treat unresectable pancreatic cancer. The review is organized in logical order from the explanation of the effect, mechanism, to challenges for translation, and supported with multiple references.

I would suggest to include a separate paragraph discussing FLASH RT in the preclinical models, using mice, pigs, and cats as this information is scattered  throughout the manuscript. 

Author Response

We thank the reviewer for the positive review.  At the suggestion of the reviewer, we made a separate section for preclinical models (Now Section 4) and reviewed preclinical data from small and large animal models.

Reviewer 2 Report

The authors provided a systematic review of the potential of FLASH radiotherapy in treating pancreatic cancers. They also discussed the challenges of FLASH RT. I agree with the authors’ opinions on the hypotheses of FLASH RT in sparing normal tissues while maintaining the treatment effects in tumors. I would recommend the acceptance of its publication. I only have some minor comments.

  • Line 109. Reference [21] is from computer simulations, and the results are not validated in experiments. Please indicate in your manuscript that this work is in-silico work. If I remember correctly, references 29 and 40 were also from computer simulations.
  • Line 111 and 164, there are two repeated periods. Please delete one.
  • Line 140, please spell out ROS at its first appearance although most readers know its meaning.
  • Line 260, I would recommend the use of “sparing effect” instead of “protective effect.” I always feel the “protective” effect is confusing or misleading. Hope you agree with me.
  • Please check the authors’ affiliation. Isn’t C.M.T. a radiation oncologist in the Department of Radiation Oncology?

Author Response

The authors provided a systematic review of the potential of FLASH radiotherapy in treating pancreatic cancers. They also discussed the challenges of FLASH RT. I agree with the authors’ opinions on the hypotheses of FLASH RT in sparing normal tissues while maintaining the treatment effects in tumors. I would recommend the acceptance of its publication. I only have some minor comments.

RESPONSE:  Thank you for the kind comments and for the support of our publication

  • Line 109. Reference [21] is from computer simulations, and the results are not validated in experiments. Please indicate in your manuscript that this work is in-silico work. If I remember correctly, references 29 and 40 were also from computer simulations

RESPONSE:  We have made the requested changes

  • Line 111 and 164, there are two repeated periods. Please delete one.

RESPONSE:  We have made the requested changes

  • Line 140, please spell out ROS at its first appearance although most readers know its meaning.

RESPONSE:  We have made the requested changes

  • Line 260, I would recommend the use of “sparing effect” instead of “protective effect.” I always feel the “protective” effect is confusing or misleading. Hope you agree with me.

RESPONSE:  We have made the requested changes

  • Please check the authors’ affiliation. Isn’t C.M.T. a radiation oncologist in the Department of Radiation Oncology?

RESPONSE:  This is correct and we have added the additional affiliation.  

Reviewer 3 Report

This manuscript is a review article which focused on the FLASH radiotherapy for Pancreatic Cancer. The authors summarized the overview and mechanisms of Flash RT and FLASH effect, followed by the feasibility and challenges of FLASH RT for pancreatic cancer.

This topic will be of interest to clinicians and researchers in the field.

However, the following major and minor issues require clarification:

Minor

  1. The authors introduced animal studies and a case treated with FLASH RT in the Challenges to Translation section. I recommend that the authors summarize such animal and human studies in another section after the Mechanism of the FLASH effect section, before focusing on the topic of pancreatic cancer.
  2. (P3L140,142) Please explain an abbreviation, “ROS”.
  3. (P6L227) Please explain an abbreviation, “PDAC”.
  4. (P7L269) Please delete “pigs and cats”.

The Conclusion section seems too long. Please summarize it properly.

Author Response

This manuscript is a review article which focused on the FLASH radiotherapy for Pancreatic Cancer. The authors summarized the overview and mechanisms of Flash RT and FLASH effect, followed by the feasibility and challenges of FLASH RT for pancreatic cancer.

This topic will be of interest to clinicians and researchers in the field.

RESPONSE:  Thank you for the supporting comments

However, the following major and minor issues require clarification:

Minor

  1. The authors introduced animal studies and a case treated with FLASH RT in the Challenges to Translation section. I recommend that the authors summarize such animal and human studies in another section after the Mechanism of the FLASH effect section, before focusing on the topic of pancreatic cancer.

RESPONSE:  We made separate sections for preclinical and clinical studies as suggested.  This suggestions has clarified the review that most of the data are preclinical

  1. (P3L140,142) Please explain an abbreviation, “ROS”.

RESPONSE:  We made the requested change.

  1. (P6L227) Please explain an abbreviation, “PDAC”

RESPONSE:  We made the requested change.

  1. (P7L269) Please delete “pigs and cats”.

RESPONSE:  We made the requested change.

The Conclusion section seems too long. Please summarize it properly.

RESPONSE:  We edited the language of the conclusion, but felt that two paragraphs were appropriate for a review to help consolidate the information in the manuscript.